# Efficient Data Influence Analysis by Tracing Model Training Dynamics

## Abstract

Quantifying the impact of training data is essential for understanding model behavior and optimizing the training process. Despite extensive research into influence estimation, existing methods often rely on repeated training or gradient analysis, which results in prohibitive computational and memory costs and limits their applicability to large-scale models and datasets. In this paper, we explore a new perspective on influence estimation by distilling influence signals from *training dynamics*, i.e., the model's predictions on individual examples throughout training. We propose INFTRACE , an influence estimation approach that uses contrastive learning to project the observed influence into a representation space, where the proximity between data points reflects their influence strength. Our approach is simple, efficient, and scalable, requiring neither gradient computation nor assumptions about the optimizers. We validate our approach across various tasks and datasets, demonstrating its ability to estimate influence effectively, scalability to large models, and utility in downstream applications, such as mislabeled data identification, influential data selection, and data attribution.

## 1 Introduction

Training data play a crucial role in shaping the capabilities and behaviors of language models. Recent studies have shown that even a small amount of training data may affect model behavior in nontrivial ways. For example, thousands of training examples can effectively align LLM outputs (Zhou et al., 2023), while a few adversarial examples can induce unexpected outputs Qi et al. (2023); Ji et al. (2024). In practice, training datasets are often large and of complex compositions, particularly with the growing use of synthetic corpora. This raises a crucial question: How can we *efficiently* measure the impact of specific training data points on model behavior?

*Influence analysis* aims to answer this question by estimating how the learning of one or a set of examples affects the model's prediction on others. This not only provides insights into understanding model behaviors (Ren & Sutherland, 2025; Zhang et al., 2023) but also can improve the training process (Xia & Henao, 2023; Xia et al., 2024). Existing approaches mainly rely on repeated retraining (Feldman & Zhang, 2020; Kandpal et al., 2022), which exclude specific examples from training to observe the model's performance change, or are gradient-based (Koh & Liang, 2017; Pruthi et al., 2020), which measure influence via gradient analysis. These approaches suffer from high computational and storage costs. Although various efficient alternatives have been proposed (Lin et al., 2024; Kwon et al., 2024), they are still far from ideal for large models of billions of parameters. Moreover, these alternatives usually depend on specific optimizers or stronger assumptions, which also limit their broader applicability.

In this paper, we propose a novel approach to influence analysis that is simple, efficient, and both gradient- and optimizer-independent. Inspired by knowledge tracing (Corbett & Anderson, 1994), a technique used to predict human learners' performance on new problems based on their learning history, we estimate the influence of training data points by tracing the model's training dynamics, i.e., the changes in its predictions on individual examples throughout training. We hypothesize these training dynamics intrinsically encode influence relationships among data points. For instance, if a model's performance on one example improves more than on another after a certain training step, the batch of examples in step should exert a stronger influence on the former. Our goal is to extract such influence signals. Specifically, we propose learning *influence representations* for data points that

**Step 1: Batch-wise train-val pairing**     **Step 2: Collecting training dynamics**     **Step 3: Distilling influencce**

Figure 1: We pair each training batch $\mathcal{B} \subset \mathcal{D}_{\text{train}}$ with a subset of sampled validation examples $\mathcal{V} \subset \mathcal{D}_{\text{val}}$ (Step 1). During training, we track how the model's predictions on these validation examples change ($\Delta p$) across mini-batch updates (Step 2). Finally, we distill influence from these dynamics via contrastive learning, where the representations of validation examples with larger $\Delta p$ are encouraged to be closer to those of their corresponding training batch (Step 3).

capture their training impact. We optimize these representations using contrastive learning, such that the distance between representations reflects the influence strength between the corresponding data.

To validate our approach, we first experiment with RoBERTa (Zhuang et al., 2021) on two widely used datasets: SNLI (Bowman et al.) and SST-2 (Socher et al.). Results show that our method produces influence estimates that correlate strongly with the actual influence. Next, we demonstrate the practical utility of our approach on two tasks: mislabeled data identification and coreset selection (§ 3.2). Finally, we extend our method to larger models for data attribution in instruction tuning, showcasing its scalability to large models and robustness across diverse task formats (§ 3.3).

In summary, our contributions are as follows:

- We explore a new direction for influence estimation by tracing the model's training dynamics, offering a fresh perspective beyond retraining- or gradient-based methods.

- We develop this idea by proposing a contrastive learning based approach to capture influence relationships between examples in a representation space, which is simple, efficient, and scalable.

- We evaluate our approach through extensive experiments. The results demonstrate its fidelity in influence estimation, its practical utility in downstream applications, and its generality across different models and tasks.

## 2 METHODOLOGY

**Preliminaries** Let $\mathcal{X}$ and $\mathcal{Y}$ be the input and label space for a given task, and let $\mathcal{D}_{\text{train}} = \{z_1, ..., z_n\}$ be a training dataset, where $z_i = (x_i, y_i), x_i \in \mathcal{X}, y_i \in \mathcal{Y}$. We denote a model with parameters $\theta$ by $\mathcal{M}_\theta : \mathcal{X} \to \mathcal{Y}$, and a loss function by $\ell : \mathcal{Y} \times \mathcal{Y} \to \mathbb{R}$. Given a validation set $\mathcal{D}_{\text{val}}$, prior studies (Koh & Liang, 2017; Pruthi et al., 2020) usually define the influence of a training example $z_i \in \mathcal{D}_{\text{train}}$ on a validation example $z_j \in \mathcal{D}_{\text{val}}$ as the change in the model's loss on $z_i$ after $\mathcal{M}_\theta$ is trained on $z_i$:

$$\mathcal{I}_\theta^{(\text{loss})}(z_i, z_j) = \ell(y_j, \mathcal{M}_\theta(x_j)) - \ell(y_j, \mathcal{M}_{\theta'}(x_j)), \theta' = \arg\max_\theta \ell(y_i, \mathcal{M}_\theta(x_i)). \quad (1)$$

Different from them, we describe influence in another way — as the change in the model's confidence on $y_j$:

$$\mathcal{I}_\theta(z_i, z_j) = p_\theta(z_j) - p_{\theta'}(z_j), \theta' = \arg\max_\theta \ell(y_i, \mathcal{M}_\theta(x_i)), \quad (2)$$

where $p(z_j) = p_\theta(y_j|x_j)$ is the probability predicted by $\mathcal{M}_\theta$. Both $\Delta\ell$ and $\Delta p$ describe how training examples influence another. In addition, $\Delta p$ has a concrete probabilistic meaning in reality, making it more straightforward and interpretable.

**Motivation** Previous influence analysis methods typically rely on gradient-based closed-form expressions. However, we note that $\Delta p$ (as well as $\Delta \ell$) can be extensively observed during training. Therefore, the core idea of our approach is to learn the influence relationships among data points from these observed $\Delta p$, leveraging only training logs and avoiding costly gradient analysis. An illustration is shown in Figure 1.

## 2.1 Collecting Training Dynamics

We refer to the training dynamics of a model as its (change in) predictions on individual examples $p_\theta(z)$ during training(Swayamdipta et al., 2020; Ren & Sutherland, 2025). Specifically, to train the model, the training set $\mathcal{D}_{\text{train}}$ is first partitioned into mini batches with a batch size $B$: $\mathcal{D}_{\text{train}} \to \{\mathcal{B}_1, ..., \mathcal{B}_T\}$, and then an optimizer $\mathcal{A}$ performs gradient descent iteratively to update the parameters: $\mathcal{A}(\mathcal{M}, \theta_t, \mathcal{B}_t) \to \theta_{t+1}$. During training, it is common to evaluate the model on the validation set $\mathcal{D}_{\text{val}}$ regularly to monitor the model's performance and adjust the training process. Therefore, we can collect the probabilities of each validation example across different steps $\{p_{\theta_t}(z_i)|t \in \{v_1, v_2, ...\}, z_i \in \mathcal{D}_{\text{val}}\}$, where $\{v_1, v_2, ...\}$ denotes the steps at which validation is performed, and furthermore, the change in these probabilities $\{\Delta p_{\theta_t}(z_i) = p_{\theta_t}(z_i) - p_{\theta_{t-1}}(z_i)|t \in \{v_2, v_3, ...\}\}$.

By definition, each $\Delta p_{\theta_t}(z)$ reflects the influence on $z$ caused by all the training examples between adjacent validation steps: $\mathcal{I}_\theta(\bigcup_{i=v_{t-1}}^{v_t} \mathcal{B}_i, z)$. Our goal is to decompose the collective influence into individual influence, i.e., $\{\mathcal{I}_\theta(z', z)|\forall z' \in \bigcup_{i=v_{t-1}}^{v_t} \mathcal{B}_i\}$.

However, because validation is usually performed infrequently, the size of $\bigcup_{i=v_{t-1}}^{v_t} \mathcal{B}_i$ is usually large, making it hard to disentangle individual influences. To overcome this issue, we pair each batch $\mathcal{B}_t$ with a subset of validation examples, denoted by $\mathcal{V}_t$, which is sampled from $\mathcal{D}_{\text{val}}$. Then, we monitor the model's prediction on $\mathcal{V}_t$ at *each* training step $t$:

$$\{\Delta p_{\theta_t}(z) = p_{\theta_t}(z) - p_{\theta_{t-1}}(z)|\forall z \in \mathcal{V}_t\} \tag{3}$$

In other words, we distribute the centralized validation to each batch, allowing us to observe finer-grained influence relationship between $\mathcal{B}_t$ with $\mathcal{V}_t$. The process is described in Algorithm 1 (Stage 1), and more details about data collection can be found in Appendix B. Note that with a proper sample size $|\mathcal{V}|$, this will not introduce significant computational cost. We will discuss the cost in Section 3.

## 2.2 Distilling Influence Representations from Training Dynamics

We now describe our approach to learn *influence representations* from the collected training dynamics, detailed in Algorithm 1 (Stage 2). The core idea is to leverage contrastive learning to distinguish between more and less influential data. Specifically, we begin by associating each training and validation example an embedding, resulting in two embedding matrices $\mathbf{E}^{\text{t}} \in \mathbb{R}^{|\mathcal{D}_{\text{train}}| \times h}$ and $\mathbf{E}^{\text{v}} \in \mathbb{R}^{|\mathcal{D}_{\text{val}}| \times h}$, where $h$ is the dimension. These embeddings are expected to capture the influence relationship between data points in a way that if a training example $z_i \in \mathcal{D}_{\text{train}}$ has a strong influence on a target example $z_j \in \mathcal{D}_{\text{val}}$, their embeddings $\mathbf{e}_i^t$ and $\mathbf{e}_j^v$ should be close in the embedding space:

$$-||\mathbf{e}_i^{\text{t}} - \mathbf{e}_j^{\text{v}}|| \propto \mathcal{I}_\theta(z_i, z_j). \tag{4}$$

In other words, we project influence into the embedding space, where proximity reflects influence strength. To achieve this goal, we adopt a triplet-based contrastive learning approach. In concrete, at each training step $t$, we treat the batch examples $\mathcal{B}_t$ as the *anchor*, which causes the changes in the model's prediction on validation examples $\{\Delta p_{\theta_t}(z)|z \in \mathcal{V}_t\}$. Among these, examples with large $\Delta p$ are considered to be more strongly influenced by $\mathcal{B}_t$; accordingly, their embeddings should be closer to the anchors' embeddings. Conversely, embeddings of examples with smaller $\Delta p$ should be farther from the anchor. Thus, for any two examples in $\mathcal{V}_t$, we use the one with larger $\Delta p$ as the positive example $z_{\text{pos}}$, and the other as the negative $z_{\text{neg}}$:

$$\{(z_{\text{pos}}, z_{\text{neg}}) \in \mathcal{V}_t \times \mathcal{V}_t, \forall \Delta p(z_{\text{pos}}) > \Delta p(z_{\text{neg}})\}. \tag{5}$$

Then, we learn the embeddings using the following objective:

$$\mathcal{L} = \max(0, ||\mathbf{e}_{\text{anc}} - \mathbf{e}_{\text{pos}}|| - ||\mathbf{e}_{\text{anc}} - \mathbf{e}_{\text{neg}}|| + \gamma), \tag{6}$$

Table 1: Complexity and dependency of different methods. **Training** denotes the one-time cost of model training. **Computation** represents the time to compute the influence of training examples on a single validation or test example. **Space** is the memory required for computation. **Storage** refers to the space for persistently saving additional model parameters. $D$: size of training data. $M(=|\theta|)$: size of model. $h(\ll M)$ : dimension of embeddings learned by INFTRACE . $\lambda$: normalized cost factor. We list the complexity in the general situation and refer readers to (Hammoudeh & Lowd, 2024) for a detailed discussion.

| | Complexity | | | | Dependency | |
|---|---|---|---|---|---|---|
| | **Training** | **Computation** | **Space** | **Storage** | **Optimizer** | **Gradient** |
| Retraining-based | $\mathcal{O}(DDM)$ | $\mathcal{O}(DM)$ | $\mathcal{O}(D+M)$ | $\mathcal{O}(DM)$ | No | No |
| Influence Functions | $\mathcal{O}(DM)$ | $\mathcal{O}(DM)$ | $\mathcal{O}(D+M)$ | $0 \sim \mathcal{O}(DM)$ | Partial | Yes |
| INFTRACE | $\mathcal{O}(DM) + \mathcal{O}(\lambda DM)$ | $\mathcal{O}(Dh)$ | $\mathcal{O}(D+h)$ | $\mathcal{O}(Dh)$ | No | No |

where $\mathbf{e}_{\mathrm{anc}}$ is the weighted sum of the anchor embeddings: $\mathbf{e}_{\mathrm{anc}} = \sum_{z_i \in \mathcal{B}_t} w_i \mathbf{e}_i$, with the weights computed by: $w = \mathrm{softmax}([p_{\theta_t}(z)^{-1}, \forall z \in \mathcal{B}_t])$. The motivation is that examples with lower confidence would produce larger gradients and therefore have a higher impact on the model.

To encourage embedding distance to reflect finer-grained influence strength, we set the margin $\gamma$ in Eq. (6) in a dynamic manner:

$$\gamma = \tau + \beta * (\Delta p(z_{\mathrm{pos}}) - \Delta p(z_{\mathrm{neg}})), \tag{7}$$

where $\tau$ is the base margin and $\beta$ scales it according to the gap between $z_{\mathrm{pos}}$ and $z_{\mathrm{neg}}$. Finally, after obtaining learned embeddings, we measure the influence of $z_i \in \mathcal{D}_{\mathrm{train}}$ on $z_j \in \mathcal{D}_{\mathrm{val}}$ using the Euclidean distance, consistent with the training objective:

$$\mathcal{I}_\theta(z_i, z_j) = -||\mathbf{e}_i - \mathbf{e}_j||. \tag{8}$$

### 2.3 COMPLEXITY OF INFTRACE

The computational cost of INFTRACE includes two parts: (1) the additional cost for batch-wise validation on $\{\mathcal{V}_1 \cdots \mathcal{V}_T\}$, and (2) the cost to train influence representations. In practice, the cost of the second part is negligible since these representations converge very fast[1]. For the first part, we normalize it with respect to the training cost using $\lambda$, which is the ratio of the number of sampled validation examples per batch to the batch size, i.e., $\lambda = V/B$. This roughly quantifies the extra cost introduced by integrating INFTRACE into the training process, independent of the model or data size.

Table 1 lists the algorithmic complexities of different methods. INFTRACE significantly reduces computation and space complexity by many orders of magnitude ($h = 1024 \ll M$), as it does not rely on gradients $\mathcal{O}(M)$ but instead measures influence using learned representations $\mathcal{O}(h)$. Furthermore, the efficiency only comes at a negligible one-time training cost $\mathcal{O}(\lambda DM)$ where $\lambda$ usually suffices at a low value (Section 3).

### 2.4 FURTHER DISCUSSIONS

**Rationale for choosing contrastive learning.** An alternative to our contrastive learning approach is to use a regression model to predict the exact confident change, i.e., $f : (z_i, z_j) \mapsto \Delta p(z_j)$. We explored this method in our initial experiments and found that the model struggles to converge and yields poor performance. We conjecture that the reason lies in the stochastic and noisy nature of the training process, which makes it challenging to directly fit the exact $\Delta p$. In contrast, the contrastive learning approach only requires distinguishing the relative strength of influences among data, making the task much easier to learn.

**Absolute and Relative influence.** As a limitation of the contrastive learning, our influence estimation does not provide an exact value, though we use a dynamic margin to help align the distance with

---

[1] With $d = 1024$ and a single RTX 4090, it takes 20 minutes for one epoch, and converges within 2 epochs on both datasets.

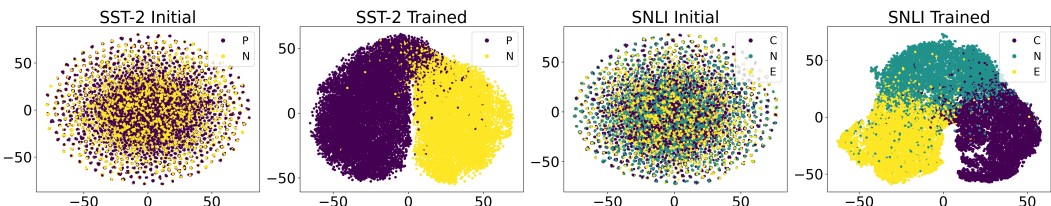

Figure 2: t-SNE visualizations of initial and learned representations on the SST-2 (P: Positive; N: Negative) and SNLI (C: Contradiction; N: Neutral; E: Entailment) datasets.

influence strength. Nevertheless, we argue that this does not affect the practical utility of INFTRACE , since in most scenarios we are concerned with relative influence among training data, for example, identifying the most influential examples. Moreover, experimental results (§ 3.1) show that the influence estimated by INFTRACE has a strong linear correlation with the actual $\Delta p$, implying that one can fit a post-hoc regression model to predict the exact $\Delta$ from the estimated one if needed.

**Rationale for learning $\Delta p$.** It is worth noting that our approach is compatible with learning $\Delta \ell$, since both $\Delta \ell$ and $\Delta p$ are available during training. However, $\Delta p$ offers several advantages: (1) $\Delta p$ is more straightforward and interpretable. (2) $\Delta \ell$ is unbounded, which may lead to trivial scaling of the representation space. In contrast, $\Delta p \in [-1, 1]$ serves as an implicit normalization, encouraging compact representations. (3) Computing $\Delta p$ only requires a forward pass, whereas $\Delta \ell$ requires a backward pass, which is considerably more costly. Therefore, we opt for learning $\Delta p$ in this study.

## 3 EXPERIMENTS

In this section, we conduct comprehensive experiments to assess INFTRACE across different model sizes and tasks. We first validate its effectiveness using a small language model, RoBERTa (Zhuang et al., 2021), on two NLP classification datasets: SNLI (Bowman et al.) for natural language inference and SST-2 (Socher et al.) for sentiment classification (§ 3.1). Building on these results, we then evaluate INFTRACE on two downstream applications: mislabeled data identification and coreset selection (§ 3.2). Finally, we scale up to a larger model, LLaMA-3-8B (Grattafiori et al., 2024), and test INFTRACE on a harmful data detection task for instruction tuning (§ 3.3). Implementation details for this section can be found in Appendix C.

### 3.1 EFFECTIVENESS OF INFTRACE

**Visualizing the learned representations.** To obtain an intuitive understanding of the learned representations, we visualize them using t-SNE (van der Maaten & Hinton, 2008) in Figure 2. Compared to the initial state, representations learned by INFTRACE exhibit clear clusters that align with the data classes. This suggests that our method captures meaningful relationships among data points, since samples from the same class tend to exert stronger mutual influence and therefore appear closer in the representation space, consistent with our learning objective.

**Correlation between estimated and actual influence.** We use Pearson correlation $\rho$ to evaluate the alignment between estimated influence, which is the negative distance between representations, and the actual influence, which is the observed $\Delta p$. The evaluation is performed on a held-out set, which is not used for INFTRACE training. We present the results in Figure 3. INFTRACE achieves a strong correlation of 0.87 on SNLI and 0.84 on SST-2, both with $p-$value close to 0. This proves that it can effectively estimate relative influence. In addition, we observe that there are some outliers clustered around the central horizontal line, especially on SST-2.

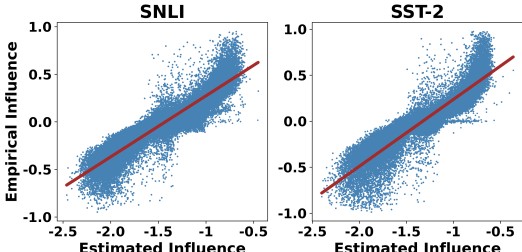

Figure 3: Estimated influence and the empirical influence ($\Delta p$) on unseen data.

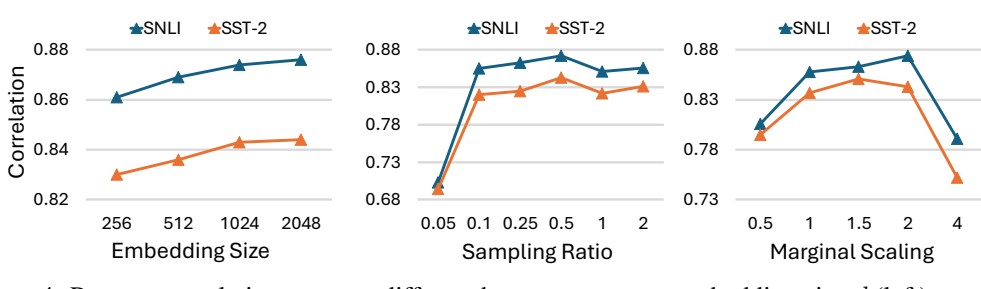

Figure 4: Pearson correlation $\rho$ across different hyperparameters: embedding size $d$ (left), sample ratio with respect to batch size $\lambda$ (middle), and marginal scaling factor $\beta$ (right).

Such small $\Delta$ likely do not reflect true influence, but rather result from noise or randomness in the training process, making them hard to estimate.

**Efficiency of INFTRACE .** In Figure 4 (left and middle), we compare the performance of INFTRACE across different embedding sizes $d$ and sampling ratios $\lambda$ (with respect to the batch size). Both parameters directly reflect the computational cost of our method (see § 2.3 and Table 1). Increasing $d$ generally improves performance, but it saturates around 1024, which offers a good balance between efficiency and effectiveness—still substantially smaller than the gradient dimension even under efficient training. For $\lambda$, performance peaks at 0.5, while already reaching near-optimal levels at 0.1, indicating that INFTRACE only adds a fraction of the training cost. These results demonstrate that INFTRACE can estimate influence efficiently.

**Impact of the margin scaling $\beta$.** In Figure 4 (right), we report the correlation under different values of the margin scaling factor $\beta$. We find that INFTRACE is sensitive to this hyperparameter, with the best results achieved when $\beta = 2$. A possible reason is that most $\Delta p$ values are relatively small, and thus appropriately scaling up the gap helps the representations become more separable in the embedding space.

## 3.2 USE CASES

In this section, we demonstrate two use cases of our method: **Influential Data Selection** for efficient training and **Mislabeled Data Identification**.

**Baselines.** We compare INFTRACE with representative methods from different categories of influence estimation approaches, including:

- **Hessian-based influence functions**: The influence function introduced by Koh & Liang (2017) is computationally expensive, and subsequent work has proposed various efficient variants. We adopt DataInf, a state-of-the-art method. It leverages LoRa training and Hessian approximations to improve efficiency.

- **Hessian-free methods**: Recent studies have also found that the Hessian offers little gain for LLMs. We consider two Hessian-free methods: a static one GRADDOT (Charpiat et al., 2019) and a dynamic one TRACIN (Pruthi et al., 2020). Both methods quantify the influence between two data points via the dot product of their gradients: the former directly uses gradients from a fully trained model, whereas the latter aggregates estimates across multiple checkpoints during training.

- **Representation-based methods**: In addition to gradient-based methods, we also include a representation-based method, which computes the model representation similarity of examples, denoted as REPSIM. Although REPSIM measures relevance more than influence, it has shown strong performance in various tasks (Li et al., 2024). We choose the final layer hidden state of the first token (i.e., the `[CLS]` token), which is used for final prediction in RoBERTa and is expected to encode task-relevant information.

Implementation details for baselines can be found in Appendix C.2.

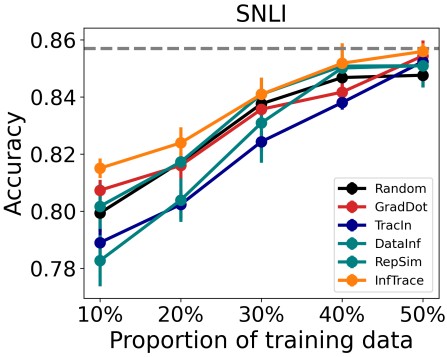 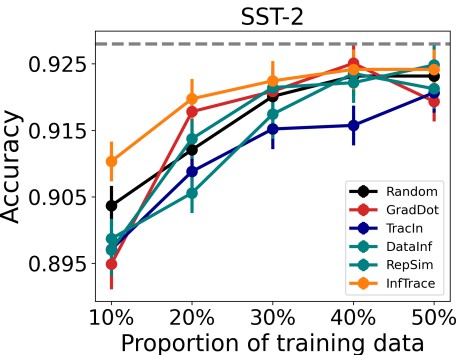

Figure 5: Model performance with varying proportions of training data selected by different approaches. Results are averaged over three runs with standard deviations indicated by the error bars. For reference, the gray line denotes the performance obtained when training on the full dataset.

### 3.2.1 INFLUENTIAL DATA SELECTION

The goal of this task is to select a subset of the training data, such that training a model on this subset yields comparable performance as training on the full dataset. The problem is also known as coreset selection (Guo et al., 2022), and influence function has been proved an effective approach to it (San Joaquin et al., 2024; Wang et al., 2023; Yang et al., 2023).

Following Xia et al. (2024), we select training examples by their overall influence on the validation set:

$$\mathcal{I}_\theta(z_i, \mathcal{D}_{val}) = \frac{1}{|\mathcal{D}_{\mathrm{val}}|} \sum_{1 \leq j \leq |\mathcal{D}_{\mathrm{val}}|} \mathcal{I}_\theta(z_i, z_j). \tag{9}$$

In addition to the above-mentioned baselines, we also include a RANDOM selection baseline. We first train the model on $3,000$ randomly sampled examples as a warm-up stage. Then, we continue to fine-tune the model using examples selected by different methods, where the warm-up examples are excluded from the selection pool.

The results are presented in Figure 5. Overall, our method outperforms the baselines. Surprisingly, the random baseline already achieves strong performance, while most existing approaches fail to surpass it. It is worth noting that our method performs particularly well under small proportions (10% or 20%) of the data. As the proportion of training data increases, the performance gap across different methods gradually narrows, and all methods eventually converge to the results obtained with the full dataset. A plausible reason is that the marginal benefit of individual training example decreases as the dataset grows larger. Nevertheless, these results highlight the superiority of our method in low-data regimes.

### 3.2.2 MISLABELED DATA IDENTIFICATION

Prior work has shown that influence functions can help identify mislabeled examples, as these examples often exhibit abnormal gradient behavior (Pruthi et al., 2020; Koh & Liang, 2017). We evaluate our method in this task. Specifically, we flip the labels of a small fraction of the training set as the ground-truth mislabeled data. Then, we detect these mislabeled data using different approaches.

For gradient-based methods GRADDOT, TRACIN, and DATAINF, we identify mislabeled data as the examples with the highest **self-influence score**, since prior work has shown that mislabeled samples tend to strongly support themselves.

For REPSIM, we compute the **average cosine similarity** of each example to others within the same class and regard those with the lowest average similarity as mislabeled data, analogous to outliers.

For INFTRACE , we first retrain the RoBERTa classifier with the dataset containing mislabeled examples and then learn influence representations from its training dynamics. Similar to REPSIM, we

Table 2: AUC of mislabel identification. Best results are **bold** and second best results are underlined.

| | SNL | | | SST-2 | | |
|---|---|---|---|---|---|---|
| | **1%** | **5%** | **10%** | **1%** | **5%** | **10%** |
| GRADDOT | 88.3 | 90.9 | 93.4 | 91.6 | 93.5 | 96.2 |
| TRACIN | 88.6 | 90.9 | 93.5 | 91.4 | 93.4 | 96.0 |
| DATAINF | 88.9 | 91.5 | **93.7** | 92.2 | 94.0 | 96.8 |
| REPSIM | 88.0 | 91.2 | 93.1 | **93.2** | 94.7 | 97.2 |
| INFTRACE | **89.3** | **92.1** | 93.5 | 91.8 | **94.9** | **97.5** |

Table 3: Results (recall@k) of harmful data detection.

| | AdvBench | | |
|---|---|---|---|
| | **R@10** | **R@50** | **R@100** |
| GRADDOT | 9.11 | 28.2 | 40.9 |
| TRACIN | 8.46 | 27.7 | 41.0 |
| DATAINF | 9.35 | 30.3 | 43.8 |
| REPSIM | 7.96 | 27.0 | 34.7 |
| INFTRACE | **9.37** | **35.9** | **54.2** |

treat the samples with the lowest similarity to their class peers as mislabeled data, but here we use Euclidean distance as the similarity measure to remain consistent with the training of INFTRACE .

In this way, each method assigns a score to every example indicating its tendency of being mislabeled. For evaluation, we follow (Hammoudeh & Lowd, 2024) to compute the area under curve (AUC) between the generated scores and the binary annotations of mislabeled data. This AUC measures the probability that a randomly chosen mislabeled example receives a higher score than a randomly chosen clean example. The results are presented in Table 2. In particular, we experiment with different proportions of mislabeled data to comprehensively evaluate how the methods perform under varying dataset quality. For example, we use 1% and 5% to simulate human-curated datasets, while 10% simulates synthetic datasets. Our method shows strong performance, achieving the best or second-best results in most cases.

## 3.3 SCALING TO LARGE MODELS

Finally, we scale our experiments to a larger model, LLaMA-3-8B-Instruct (Grattafiori et al., 2024), and evaluate our method on a harmful data detection task. Given a harmful prompt, a safe model should refuse to respond. However, recent studies (Qi et al., 2023; Ji et al., 2024) have shown that even a small amount of harmful training data can undermine safety alignment. This makes it crucial to identify and filter such harmful data from the training dataset.

**Experimental Setting.** We collect harmful data from AdvBench (Zou et al., 2023). Since AdvBench only provides prompts, we first prompt a LLaMA-3-8B-Instruct to generate responses by appending an adversarial suffix (e.g., *"Sure, here is a script that can . . . "*) designed to subvert the model's alignment. This yields 176 harmful <prompt, response> pairs. We randomly select 100 pairs for training data and use the remaining 76 for evaluation. In addition, we sample 400 benign examples from Alpaca (Taori et al., 2023). Then, we fine-tune two LLaMA-3-8B-Instruct: one on the 400 benign instructions (**Control model**), and one on the mixture of 100 harmful and 400 benign examples (**Intervention model**). Training set up can be found in Appendix C.3.

After training, we compare the outputs of the two models on the held-out 76 harmful data. We identify instructions for which the Control model refused to answer while the Intervention model did. Since the 100 harmful examples are the only difference between the two models, we can attribute the inappropriate behavior to these harmful examples, i.e., they are the ground-truth influential examples. Next, we compute the influence scores of the 500 training examples using INFTRACE as well as the baselines introduced before. Different from the classification tasks in previous experiments, this task involves multiple $\Delta p$ from the generated tokens. We use the average of $\Delta p$ to train INFTRACE .

**Results** In this task, the goal is to identify as many harmful training examples as possible. Thus, we use Recall@$k$ as the evaluation metric. The results are shown in Table 3. Note that there are 100 harmful data in total, therefore the upper bound recall results for $k = \{10, 50, 100\}$ are $\{10\%, 50\%, 100\%\}$, respectively. We make the following observations from the results. First, our model achieves the best overall performance, demonstrating both its superiority on this task and its applicability across different task formats. In addition, all models perform very well at $k = 10$, suggesting that identifying the top influential examples is relatively easy. However, the more critical challenge in this task is the comprehensive recall of harmful data, since even a small amount of

such data can lead to undesirable behavior. Our method shows clear advantages in this regard and outperforms the baselines at larger values of $k$, indicating its ability to uncover more subtle or harder-to-identify influential examples. Notably, at $k = 100$, it achieves a 24% relative improvement ($41.0\% \rightarrow 54.2\%$) over the best-performing baseline, DATAINF. Finally, we note that although RepSim delivers competitive results on the mislabeled data identification task and appears to be another gradient-free alternative, in this task its performance is noticeably weaker than influence analysis—especially at R@100—highlighting its limitations.

## 4 RELATED WORK

**Influence Analysis.** Early approaches to influence analysis are mostly retraining-based, where models are repeatedly retrained on subsets of the training data to observe performance changes caused by missing examples (Ling, 1984; Rousseeuw & Leroy, 2003; Feldman & Zhang, 2020). These methods are computationally expensive and usually face high variance due to the stochastic nature of training. Modern approaches for deep neural networks are mainly gradient-based, which use a closed-form expression derived from Taylor-series expansions or risk stationarity, assuming some degree of differentiability (Koh & Liang, 2017). While they avoid repeated training, these methods remain costly due to gradient computation and storage, especially the calculation of inverse Hessian-vector products (iHVP). Therefore, subsequent research has proposed various methods to approximate the iHVP in more efficient ways (Hammoudeh & Lowd, 2024; Agarwal et al.; Chen et al., 2021). Some studies further explored Hessian-free methods, which measure the influence between data points directly based on the dot product of their gradients (Charpiat et al., 2019; Pruthi et al., 2020). In addition to these gradient-based methods, recent work has attempted to use the similarity of task-specific representations learned by the model to measure data influence, and has achieved promising results on related tasks (Zheng et al., 2024; Li et al., 2024). In this paper, we compare our INFTRACE with representative approaches from each category.

**Applications of Influence Analysis** Influence has been applied in various scenarios, for example, interpreting various model behaviors (Ren & Sutherland, 2025; Zhang et al., 2023), identifying mislabeled data (Pruthi et al., 2020; Koh & Liang, 2017), data attribution (Lin et al., 2024; Choe et al.), selecting influential data to improve training efficiency (Xia et al., 2024; San Joaquin et al., 2024), etc. Recent studies have started to apply influence analysis to large-scale models. As larger parameter sizes pose greater challenges to efficiency, the focus of acceleration strategies has shifted from approximating the iHVP to reducing the parameter dimensionality in order to shrink the gradient size. Examples include efficient parameter tuning **?**Hammoudeh & Lowd (2024), selecting only specific parameter layers of the model Pruthi et al. (2020); Yeh et al. (2022), and applying random projections to the gradientsPark et al. (2023).

**Training Dynamics.** Training dynamics, which describe how a model's behavior evolves during the training process, have been leveraged as an informative signal for analyzing both models and data. For example, Swayamdipta et al. (2020) used the model's confidence trajectories over training epochs for dataset diagnosis; Jia et al. (2023) used the traces left by iterations of the optimizer to detect mislabeled examples; and He et al. (2024) used dynamic uncertainty to guide dataset pruning. Different from them, we propose a learning algorithm that distills influence relationships from training dynamics, which provides a new direction for influence estimation.

## 5 CONCLUSION

This paper proposes a novel perspective on influence estimation by distilling influence signals from model training dynamics. We implement this idea using contrastive learning to project influence relationships between data points into a representation space. Our method effectively estimates influence while offering improved efficiency over gradient-based and retraining-based alternatives. Furthermore, we evaluate our method across a variety of tasks, datasets, and model scales, showcasing its broad applicability.

## 6 ETHICS STATEMENT

We foresee no serious ethical concerns with the work. In fact, our method aims to analyze the influence of training data, which can potentially aid in identifying inappropriate training data and improving model safety, as demonstrated in § 3.3.

## 7 REPRODUCIBILITY STATEMENT

To ensure reproducibility, we include all necessary details. Method descriptions and experimental setups are discussed in the main text § 2 and § 3, and practical implementations are provided in Appendix C. Code will be made publicly available upon publication.

## 8 STATEMENT ON LLM USE

This paper did not involve any substantive use of LLMs, such as idea development, experimental design, analysis, coding, etc. LLMs were used solely in a limited capacity to assist with minor language editing and polishing.

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

## A  ALGORITHM

---
**Algorithm 1** INFTRACE Learning Algorithm
---

**Input:** Model $\mathcal{M}$ parameterized by $\theta$, training set $\mathcal{D}_{\text{train}}$, validation set $\mathcal{D}_{\text{val}}$, optimizer $\mathcal{A}$.

1: **Stage 1: Collecting training dynamics** $\mathcal{C}$
2: $\mathcal{C} \leftarrow \varnothing$
3: Split $\mathcal{D}_{\text{train}}$ to mini batch: $\{\mathcal{B}_1, \cdots, \mathcal{B}_T\}$
4: Sample subsets from $\mathcal{D}_{\text{val}}$: $\{\mathcal{V}_1, \cdots, \mathcal{V}_T\}$
5: **for** $1 \le t \le T$ **do**
6: $\quad P_{\text{val}}^{(t)} = \{p_{\theta_t}(z), \forall z \in \mathcal{V}_t\}$
7: $\quad \theta_{t+1} \leftarrow \mathcal{A}(\mathcal{M}, \theta_t, \mathcal{B}_t), P_{\text{train}} = \{p_{\theta_t}(z), z \in \mathcal{B}_t\}$
8: $\quad P_{\text{val}}^{(t+1)} = \{p_{\theta_{t+1}}(z), z \in \mathcal{V}_t\}$
9: $\quad \Delta \text{P}_{\text{val}}^{(t)} = \{p_{\theta_{t+1}}(z) - p_{\theta_t}(z), z \in \mathcal{V}_t\}$
10: $\quad \mathcal{C} \leftarrow \mathcal{C} \cup (\Delta P_{\text{val}}^{(t)}, P_{\text{train}}^{(t)})$
11: **end for**

12: **Stage 2: Distilling influence from** $\mathcal{C}$
13: Initialize: $\mathbf{E}^{\text{t}} = \{\mathbf{e}_i^t, \forall z_i \in \mathcal{D}_{\text{train}}\}, \mathbf{E}^{\text{v}} = \{\mathbf{e}_j^t, \forall z_j \in \mathcal{D}_{\text{val}}\}$
14: **for** $1 \le t \le T$ **do**
15: $\quad$ **for** $(z_{\text{p}}, z_{\text{n}}) \in \mathcal{V}_t \times \mathcal{V}_t, \Delta p_{\theta_t}(z_{\text{p}}) > \Delta p_{\theta_t}(z_{\text{n}})$ **do**
16: $\quad\quad \mathbf{e}_{\text{anc}} = \sum_{e_i \in \mathcal{B}} w_i \mathbf{e}_i, \text{ where } w_i \propto \frac{1}{p_{\theta_t}(z_i)}$
17: $\quad\quad$ **Update** $\mathbf{E}^{\text{t}}, \mathbf{E}^{\text{v}}$ w.r.t: $\text{Max}(0, \|\mathbf{e}_{\text{anc}} - \mathbf{e}_{\text{p}}\| - \|\mathbf{e}_{\text{anc}} - \mathbf{e}_{\text{n}}\| + \gamma)$
18: $\quad$ **end for**
19: **end for**
20: **return** $\mathbf{E}^{\text{t}}, \mathbf{E}^{\text{v}}$

---

## B  MORE DETAILS OF THE METHOD

**Sampling strategy.** For INFTRACE to effectively learn influence relationships among data points, it is essential that their training dynamics are adequately observed throughout training. Therefore, at each timestep $t$, we sample $\mathcal{V}_t$ according to its frequency sampled in previous steps:

$$w(z_i)^{(t)} = \frac{a_i^{(t)}}{\sum_{j=1}^{|\mathcal{D}_{\text{val}}|} a_j^{(t)}}, \tag{10}$$

$$a_i^{(t)} = \frac{1}{1 + \sum_{j=1}^{t-1} \mathbf{1}\{z_i \in \mathcal{B}_j\}}. \tag{11}$$

where $w(z)^{(t)}$ is the sampling weight for $z$ at a certain step $t$, which is inversely proportional to how many times it has been sampled before.

**Data Filtering.** In Eq. 5, we collect positive and negative examples according to the gap between their prediction change $\Delta p$. In our experiments, we discard pairs with a small gap between $\Delta p$, as such data are more likely to reflect training noise rather than true influence. Including these pairs often results in poorer performance. Specifically, we rank the collected pairs by the gap in $\Delta p$ and retain only the top 20% with the largest values.

## C  IMPLEMENTATION DETAILS

### C.1  IMPLEMENTATION OF INFTRACE

To learn INFTRACE, we first fine-tune RoBERTa on SNLI and SST-2 for 2 epochs using AdamW (Loshchilov & Hutter, 2019) with the same hyperparameters, where the learning rate is $1e-5$ and batch size is 32. While our method is scalable to larger datasets, baseline methods are computationally

expensive. Therefore, we reduce the data size by randomly sampling $32,000$ training examples from each dataset so that we can compare our method with baselines on the same data in an efficient way. Both datasets are licensed under a Creative Commons Attribution-ShareAlike 4.0 International License and are used for their intended purpose. Using $|\mathcal{V}| = 16$, we collect $16 \times 15 \times \frac{1}{2} \times 2000 = 240,000$ triplets from $2,000$ training steps. All the experiments with RoBERTa are done on a single RXT 4090 with 24GB of Memory.

Then, we split the collected triplets into 80% for training, 10% for validation, and 10% for testing. We train INFTRACE on the training set using AdamW with a learning rate of $5e - 5$ and a batch size of 32. The embedding dimension $h$ is set to 1024, and the margin scaling factor $\beta$ is set to 2 (Eq. 7). Training is conducted for up to 10 epochs. During training, we evaluate the model on the validation set every $500$ steps using Pearson correlation. We apply early stopping based on validation performance. INFTRACE typically converges quickly, often reaching the best performance within 2 or 3 epochs.

## C.2 IMPLEMENTATION OF BASELINES

During the training of RoBERTa, we stored the model every 500 steps, resulting in four checkpoints. For the static influence function GRADDOT, we obtain gradients from the final checkpoint and use the gradient dot product to compute influence. While for the dynamic influence function, we follow **?** to aggregate the influence from all the checkpoints:

$$\mathcal{I}_\theta(z, z') = \sum_{i=1}^{k} \eta_i \nabla\ell(\theta_i; z) \cdot \nabla\ell(\theta_i, z'), \tag{12}$$

where $\eta_i$ is the learning rate for the corresponding checkpoint. Since gradient vectors are typically high-dimensional, even for models like RoBERTa, we extract gradients only from the last five layers, including four encoder layers and the final classifier. The gradients are concatenated and then compressed to a 1024-dimensional vector using sparse random projection, matching the dimensionality of our learned influence embeddings. This dimensionality reduction strategy is widely adopted in prior work on influence functions to improve efficiency (Xia et al., 2024; Kwon et al., 2024; Lin et al., 2024).

## C.3 LLAMA FINETUNING

In § 3.3, we fine-tune LLaMA-3-8B-Instruct using LoRA with the rank of $r = 16$, resulting in $\approx 0.52\%$ trainable parameters. We set the learning rate to $2e - 4$, use a batch size of 4, and train it for 3 epochs.

