# OpenReview forum: "Efficient Data Influence Analysis by Tracing Model Training Dynamics"
_ICLR.cc/2026/Conference — ICLR 2026 Conference Withdrawn Submission_

### Official Review · Reviewer_HFmc · 2025-10-14

**Soundness:** 2
**Presentation:** 2
**Contribution:** 2
**Rating:** 2
**Confidence:** 5

**Summary:**

The paper proposes INFTRACE, a gradient-free algorithm for estimating data influence by distilling training dynamics into a learned embedding space. Instead of relying on gradients or Hessians, it tracks how each batch affects the prediction of validation samples during training and then learns a contrastive embedding where positively influenced validation samples lie closer to the training batch. Influence between any training and validation pair is computed as the negative distance between their learned embeddings. The authors demonstrate the effectiveness of their approach on diverse models (RoBERTa and LLaMA-3-8B) and tasks (mislabeled data detection, data selection).

**Strengths:**

- INFTRACE offers a gradient-free alternative to influence estimation that avoids costly Hessian or gradient computations, making it applicable to large models such as LLaMA-3-8B
- Comprehensive experiments across a variety of models and tasks

**Weaknesses:**

- Lack of principled formulation and interpretability:

  - INFTRACE fails to account for the sign or direction of influence both in its algorithmic design and final scores. 1) In the contrastive learning framework, validation samples with large negative $\Delta p$ values are always treated as negatives samples and therefore forced to be far away from the batch embedding. This does not look like a principled design. 2) The INFTRACE scores are negative for all training samples and therefore offers no interpretation as to which training samples are helpful or harmful. This is a significant drawback and contrasts with most existing data attribution methods.

  - INFTRACE uses a weighted sum of training embeddings instead of modeling how each individual training sample affects a validation example (as in TracIn). This aggregated representation blurs the source of influence within the batch. Moreover, the inverse-confidence weighting scheme is heuristic, lacking both theoretical justification and ablation analysis.

  - From an information-theoretical perspective, relying solely on prediction deltas ($\Delta p$) is unlikely to yield more informative or faithful influence estimates than gradient-based approaches such as TracIn, which leverage richer, first-order information.

- Gross ignorance of prior work: The paper overlooks a substantial body of recent literature on dynamic influence estimation. TracIn itself is a dynamic method that tracks influence along the training trajectory, yet this is not properly acknowledged. In addition, SGD-influence [1] and Simfluence [2] are two prominent approaches that the authors fail to cite or compare against. The paper also ignores recent efforts to scale these dynamic methods to LLMs (e.g., [3, 4]). I encourage the authors to check the recent survey on data attribution [5] for a more comprehensive view of this rapidly evolving area.

- The empirical results (in particular Figure 5 and Table 2) do not show statistically significant improvement over baselines. The authors may consider incorporating additional evaluation metrics, such as the Linear Datamodeling Score (LDS).


References

[1] Hara, Satoshi, Atsushi Nitanda, and Takanori Maehara. "Data cleansing for models trained with SGD." Advances in Neural Information Processing Systems 32 (2019).

[2] Guu, Kelvin, et al. "Simfluence: Modeling the influence of individual training examples by simulating training runs." arXiv preprint arXiv:2303.08114 (2023).

[3] Wang, Jiachen T., et al. "Data shapley in one training run." ICLR 2025

[4] Wang, Jiachen T., et al. "Capturing the Temporal Dependence of Training Data Influence." ICLR 2025

[5] Deng, Junwei, et al. "A Survey of Data Attribution: Methods, Applications, and Evaluation in the Era of Generative AI." (2025).

**Questions:**

I don’t have further questions for now. My primary concerns lie in the design principles of INFTRACE and the lack of comparison with recent, strong baselines (e.g., [3] and [4]).

---

> ### Author Response · Authors · 2025-11-20
>
> Thanks for your review. Please see our response below.
>
> ---
>
> **Response to Weakness 1: Lack of principled formulation and interpretability**
>
> - **Response to "INFTRACE fails to account for the sign or direction of influence both in its algorithmic design and final scores."**
>
> This is factually incorrect and a misunderstanding of our method. InfTrace is trained to predict $\Delta p$, **which falls into $[-1, 1]$ and has the sign and direction**: $>0$ means helpful and <0 means harmful.
> Although the final scores (computed as negative distance) are always negative, they are highly correlated to the actual influence (See Figure 3). **This means that there will be a (negative) value that corresponds to zero influence and separates the positive and negative influential examples.** This is reflected in Figure 3, where the scores clearly capture both directions of influence ($\Delta p<0$ or $\Delta p >0$).
>
> - **Response to "The design of InfTrace involves heuristics"**
>
> Heuristics are common in AI method design. Take TracIn as an example, it uniformly distributes a batch’s influence across all examples, which is also a heuristic approximation.  Moreover, its use of checkpoints is explicitly described as a heuristic (the title Section 3.3 of the original paper). We believe that the appropriate criterion is whether the design works in practice. Our experimental results demonstrate that our approach is effective.
>
> - **Response to whether $\Delta p$ is informative of influence.**
>
> First, in our method, influence is defined as $\Delta p$, and is therefore naturally informative by definition.
> Beyond this, we want to explain why our way of collecting $\Delta p$ is effective by comparing it to leave-one-out (LOO) training.
>
> Conceptually, LOO and our collection procedure are similar: both collect empirical influence by observing how model outputs change across repeated training runs: If we view each minibatch update as one “training run”. The differences lie in:
>
> 1. LOO focuses on the effect of excluded points, while ours focuses on the effect of included points.
> 2. Ours is extremely sparse in terms of the number of examples per train (minibatch). Therefore, it cannot readily obtain all the influence like LOO. This motivates InfTrace which learns to predict unobserved $\Delta p$ from limited observed ones.
>
> In other words, both our approach and LOO are empirical methods, whereas influence functions are theoretical. Both approaches should be able to yield effective influence estimates when applied appropriately, and our experimental results proved this.
>
> ---
>
> **Response to Weakness 2: Ignorance of recent literature on “dynamic” methods**
>
> We would like to clarify that the notion of “dynamic” in the suggested papers is fundamentally different from the “dynamics” in our paper. In our paper, the term "training dynamics" refers specifically to the evolution of $\Delta p$ during training (Line 116-117). By contrast, the suggested papers, such as TracIn or “Capturing the Temporal Dependence of Training Data Influence”, their “dynamic” means taking into account the influence at different training stages. The fundamental difference is that these methods remain gradient-based, while our method solely leverages $\Delta p$.
>
> Nevertheless, we are happy to include the suggested papers in the revised version.
>
> ---
>
> **Response to Weakness 3: The improvements are not significant (Fig 5, Tab 2)**
>
> - First of all, we argue that **the results should be interpreted from a holistic perspective**. Our evaluation spans multiple downstream tasks and covers both classification and generation settings. InfTrace achieves the best overall performance across these scenarios, showing both its applicability and robustness.
>
> - Second, when scaling to larger models (Table 3), InfTrace outperforms baselines by a larger margin. This is particularly important given the prevalence of LLMs and generation tasks.
>
> - Finally, we believe the results should be viewed alongside the goal and intended contributions. Our goal is not to propose a state-of-the-art method that dramatically outperforms existing ones.
>
> Instead, our contributions are:
> 1. Offering a fresh, gradient-free perspective on influence analysis and pointing to an underexplored direction for fuure research;
> 2. Empirically showing that this perspective yields a strong and efficient alternative to existing widely used methods.
>
> ---
>
> **Response to "Some baselines are not compared against"**
>
> Consistent with the intended contributions of the paper, our baseline selection prioritizes diversity and representativeness rather than competing with the latest SOTA methods. We therefore select **typical and widely-adopted representatives from each major category of influence analysis**—gradient-based (Hessian-based, Hessian-free, static, dynamic), and representation-based. The goal is to show it can be a strong yet efficient alternative.

---

> > ### Comment · Reviewer_HFmc · 2025-11-20
> >
> > The authors selectively responded to my comments. I will clarify a few points and close the discussion.
> >
> > - Even assuming the existence of "a negative value that corresponds to zero influence and separates the positive and negative influential examples", there is no easy way of knowing what this value is without computing the actual influence. This creates burden for interpretability.
> >
> > - The interpretation of the paper "Capturing the Temporal Dependence of Training Data Influence" is wrong. This paper, along with the paper that it builds upon, SGD influence, are all about modeling the dynamics of training trajectory when removing a single sample. While they can be used to compute the influence at different training stages, these papers are fundamentally about how LOO influence evolves over training.

---

> > > ### Author Response · Authors · 2025-11-21
> > >
> > > We respectfully ask you to clarify which part of your comments we did not respond to, and we are happy to address it.
> > >
> > > - We can easily obtain the approximate separating point from a held-out set, as shown in Figure 3. This held-out set is split from the collected $\Delta p$ and does not require any additional computation. Could you elaborate on why *"it creates burden for interpretability?"*
> > >
> > > - Could you please specify which part of our interpretation of *“Capturing the Temporal Dependence of Training Data Influence”* you believe is incorrect? To the best of our understanding, this paper remains a gradient-based method, and our method is gradient-free, which is a fundamental methodological difference—and this is the point we are making.

---

### Official Review · Reviewer_bCCZ · 2025-10-28

**Soundness:** 2
**Presentation:** 3
**Contribution:** 2
**Rating:** 4
**Confidence:** 3

**Summary:**

This paper introduces INFTRACE, an efficient method for data influence analysis that avoids the high costs of traditional gradient-based or retraining-based approaches. The core idea is to distill influence signals from training dynamics. INFTRACE then uses contrastive learning to map data points into a low-dimensional representation space, where the proximity between two points directly reflects the influence strength between them. The authors demonstrate that this gradient-free and optimizer-independent method is scalable and effective on models like RoBERTa and LLaMA-3-8B for tasks including mislabeled data detection and influential data selection

**Strengths:**

The idea of using contrastive learning to learn a dedicated "influence representation space" is both interesting and novel. This "influence embedding" perspective is a fresh contribution to the field.

The paper is well-motivated, clearly identifying the computational and storage bottlenecks of existing methods. The proposed INFTRACE, being gradient-free and optimizer-independent, directly addresses this "efficiency" motivation, which is supported by the complexity analysis.

**Weaknesses:**

The primary validation in Section 3.1 (Fig. 3) relies on correlating the estimated influence with the empirical $\Delta p$. This is, to some extent, circular reasoning, as the model was explicitly trained to capture signals from $\Delta p$. A more robust validation would compare the estimated influence to a "ground truth" measure of influence, such as the actual change in model parameters or loss from Leave-one-out (LOO) retraining, even if only on a small-scale setup.

The method appears to require re-training for each specific task, dataset, and model. The learned embeddings are specific to the training dynamics of a particular run. This limits the method's generalizability, as it cannot be readily applied to a new, pre-trained model or dataset without re-collecting all training dynamics and re-training the contrastive model.

The accuracy of INFTRACE is dependent on the quality of the learned representations, which in turn depends on the amount of data (i.e., the number of triplets) used to train the contrastive learning model. While the paper ablates the sampling ratio $\lambda$ (Fig. 4, middle), it does not explicitly discuss how the total number of collected triplets affects the final accuracy of the influence estimation. This is a key parameter for practical application.

**Questions:**

1.Could the authors elaborate on the feasibility of comparing INFTRACE to Leave-one-out (LOO) on a smaller-scale experiment? This would significantly strengthen the claim that the method captures "true" influence rather than just its own proxy metric ($\Delta p$).

2.Could the authors provide an experiment or discussion on the impact of the contrastive learning dataset size on the accuracy of the influence estimation? This would provide practical guidance on how much data need to be collected for the embeddings to stabilize.

3.The paper notes that REPSIM performs poorly on the LLaMA task, while INFTRACE excels (Table 3), especially at R@100. Could the authors elaborate on why this is the case?

4.The choice of $\Delta p$ (change in probability of the correct class) is clear for classification tasks. However, for continuous, auto-regressive generation tasks (like with LLaMA), how is $\Delta p$ precisely defined?  Is it the average change in log-probability for all tokens in the generated sequence, or the change in probability of a specific target class?

5.The rationale for choosing $\Delta p$ over $\Delta l$ is given in Sec 2.4 (lines 240-248), arguing it is more interpretable, bounded, and computationally cheaper. However, could the authors provide experimental evidence? For example, what happens if INFTRACE is trained to model the gap in $\Delta l$ instead of $\Delta p$? Would its performance on downstream tasks degrade?

---

> ### Author Response · Authors · 2025-11-20
> **Responses to Weaknesses**
>
> Thanks for your helpful feedback. Please see our responses below.
>
> ---
>
> **Response to "Validation in Section 3.2 appears to be circular"**
>
> We believe this is not circular. Although the training and evaluation are both based on $\Delta p$, **we split the collected $\Delta p$ into a training set and a test set (Line 263-265), and the evaluation results in Section 3.2 are based on unseen data**, which means it evaluates whether representations learned from observed $\Delta p$ is generalizable to predict unobserved $\Delta p$.
>
> This setup is not circular: if we followed that logic, all supervised learning would be “circular” simply because models are trained on labels $Y$ and also evaluated on labels $Y$. As long as the training and test sets are separated, the evaluation measures generalization rather than reusing the same evidence to justify itself.
> In fact, this alignment between how the representations are trained and how they are used is an advantage of our method design.
>
> ---
>
> **Response to "InfTrace requires re-training for a new task/model/dataset"**
>
> We acknowledge this. In practice, this increases the cost factor $\lambda$ to $1+\lambda$.
> However, compared to LOO that requires $D$ times training and gradient-based methods that require vast amount of gradient computation, InfTrace remains relatively efficient (See the computation complexity in Tab 1).
>
> Besides, there might be some solutions, for example, instead of learning influence representations from scratch, one can learn a function that maps certain hidden layers of the model to influence representations. Leveraging $\Delta p$ to estimate influence is an underexplored direction, and we leave this to future work.
>
> ---
>
> **Response to "the learned representations only apply to one particular run"**
>
> This is actually a common issue in influence estimation and has also been discussed in prior work [1]. It is largely caused by the inherent randomness of training.
> However, in practice, such variances are usually limited and do not affect the effectiveness of the influence functions. The same holds for ours. This is reflected in its performance on downstream tasks. For example, in the coreset selection task, we select data using one run, then retrain the model multiple times using the selected data. Still, these runs yield strong coreset selection performance, showing that the influential data selected in one run are also influential in other runs.
>
> Similarly, in the mislabeled data identification task, our method (as well as baselines) can accurately detect mislabeled data within a single run. This shows that the noise does not overwhelm the signal.
>
> [1] Trak: attributing model behavior at scale
>
> ---
>
> **Response to "the absolute amount of data required for training InfTrace**
>
> The amount of data required depends on the size of the training set, since our goal is to learn the influence of every training example. As the number of training examples increases, more triplets are needed. Therefore, we introduce the sampling ratio $\lambda$ as a normalized factor to quantify how many triplets are needed, which we believe is more meaningful and informative to provide practical guidance: one can decide the absolute sampling (and hence the amount of triplets) according to their training data size.

---

> ### Author Response · Authors · 2025-11-21
> **Responses to Questions**
>
> **Feasibility of comparing InfTrace to LOO**
>
> LOO is very expensive and, more importantly, is not always considered ground-truth because it’s also prone to the noise of a particular run. Prior studies in influence analysis also typically do not compare against LOO.
> Besides, the essence of LOO is to observe empirical influence from repeated training runs. From this perspective, our way of collecting $\Delta p$ shares a similar idea with LOO: by treating each minibatch update as a ‘training run,’ we also obtain $\Delta p$ from a large number of repeated empirical training outcomes.”
>
> Nevertheless, we will try to perform a small-scale LOO and include it in the revised version if computational resources permit.
>
> ---
>
> **The impact of contrastive data size**
>
> Please see our response to weaknesses.
>
> ---
>
> **Why RepSim performs poorly on LLaMA while ours performs well**
>
> Similarity and influence are fundamentally different concepts: RepSim mainly captures the former, whereas our representations are explicitly optimized to learn the latter. Therefore, it is expected that our method performs better.
>
> As for why our method surpasses RepSim by a larger margin when k is larger, we hypothesize that semantically similar examples also tend to exert stronger influence on each other. Consequently, when *k* is small, RepSim retrieves top semantically similar examples that may overlap more with influential ones.
>
> ---
>
> **How to use Δp\Delta pΔp in generation tasks**
>
> As described in Lines 422–423, we simply used the average of token-level $\Delta p$.
>
> ---
>
> **Comparision between $\Delta p$ and $\Delta l$**
>
> $\Delta p$ and $\Delta l$ are mathematically related, so one is unlikely to provide substantially more information than the other. However, from a feasibility perspective, the bounded nature of $\Delta p$ should make it easier and more suitable for contrastive learning.
> Therefore, we choose $\Delta p$. We will include a comparison in the revised version. Nevertheless, this is an implementation detail and does not affect the validity of our method.

---

> > ### Comment · Reviewer_bCCZ · 2025-11-26
> >
> > Thank you for your response. Some of my concerns have been addressed. However, after reading the other reviewers’ comments, I found that I share some of the same concerns, so I have decided to maintain my score.

---

### Official Review · Reviewer_8mWJ · 2025-10-31

**Soundness:** 2
**Presentation:** 2
**Contribution:** 2
**Rating:** 2
**Confidence:** 3

**Summary:**

This paper proposes INFTRACE, an approach to estimate data influence by observing model training dynamics. The method collects the change in prediction probabilities on validation examples throughout the training process. It then uses a contrastive learning objective to learn an embedding space where the proximity between data points is intended to reflect their influence strength. The utility of this method is evaluated on tasks like mislabeled data identification and influential data selection.

While the work is based on a simple and reasonable design, the central claims of efficiency are not supported and appear to be incorrect. The method introduces a heavy computational overhead, the complexity analysis in Table 1 is flawed and incomplete, the empirical evaluations are weak, and the results are not significant enough to justify the costs. The paper does not have enough contribution for a full publication. The authors should consider expanding the scope, addressing the severe computational costs, and re-evaluating the method for a workshop or as a short paper submission.

**Strengths:**

1. The paper presents a novel perspective on influence estimation by
evaluating on validation examples at each training step rather than relying on standard gradient-based or retraining methods.

2. The method design is straightforward and intuitive, and it could be useful in specific practical use cases where full training process is accessible.

**Weaknesses:**

1. The paper's title and abstract claim the method is "efficient." However, the proposed approach appears to be significantly more computation-heavy than many existing methods. INFTRACE requires consistently evaluating the model on validation examples at every training step to collect probability changes. This introduces a very high computational overhead that is not adequately acknowledged.

2. The complexity analysis in Table 1 is unclear, incomplete, and misleading. It fails to properly account for the high cost of INFTRACE's own data collection phase (i.e., step-by-step validation), which is a critical part of its training cost. The table is missing many important baselines, such as TracIn, [Data Shapley in One Training Run], and [The Mirrored Influence Hypothesis].


3. The empirical results are not significant. The performance differences shown in Figure 5 and Tables 2/3 are marginal. Given the significantly higher computation cost of INFTRACE and its dependency on having access to the entire training process, these slight improvements are insufficient to demonstrate advantages over existing approaches.

3. The analysis of the learned representations (Figure 2) seems potentially irrelevant. The t-SNE visualization primarily shows that the embeddings separate samples by their class labels. This can be a trivial result, as it could be achieved by simply concatenating a one-hot label encoding to the features. It is unclear what the "initial embeddings" represent (e.g., from a randomly initialized model? If so, the random distribution is trivial). Most importantly, how is this label-separation property connected o the task of capturing sample influence, which is the stated goal of the embeddings?

4. The correlation plot shows the learned representation compressed the influence relationships. However, it does not sufficiently motivate why this compression is necessary or beneficial for the task of influence estimation. Why do we need to do this?

**Questions:**

In Table 1, why do Retraining-based methods have computational complexity of $O(DDM)$?

---

> ### Author Response · Authors · 2025-11-20
> **Responses to Weaknesses**
>
> Thanks for your review. Please see our responses to the weaknesses below.
>
> ---
>
> **Response to Weakness 1: The computation of InfTrace is significantly heavier than existing methods due to the cost of extra validation.**
>
> This is not true. We thoroughly discuss this issue and **use a factor of $\lambda$ to quantify the computational overhead of extra validation, which is normalized to the training cost** (See Lines 191-196). This is also reflected in Table 1 (The training column). Furthermore, we **experimentally demonstrate that $\lambda$ can be set to a very small value (e.g., 0.1) while still achieving nearly optimal performance.** Therefore, we demonstrate the efficiency of our method both theoretically and empirically.
>
> ---
>
> **Response to Weakness 2: About the complexity analysis in Table 1**
>
> - **Response to “Table 1 fails to account for INFTRACE’s data-collection cost”**
>
> As with the previous response, we respectfully ask the reviewer to revisit Section 2.3 (Lines 190–201), where we provide a detailed analysis of the cost of step-level validation. In Lines 286–292, we also report and discuss the empirical additional cost.
>
> - **Response to “Many important baselines are missing in Table 1”**
>
> This is not true. Because there are many different influence-analysis methods, it is not feasible to list the complexity of every individual method. Instead, we follow the standard practice of grouping prior work by method category. **Note that the second row “Influence Functions” covers various gradient-based methods.**
>
> For example, TracIn mentioned by you falls into gradient-based approach and has computation complexity $O(DMT)$, where $D$ is dataset size, $M$ is the number of model parameters, and $T$ is the number of checkpoints (a constant, and therefore can also be written as $O(DM))$. For full details, we refer the reviewer to Section 5.2.1 of [1] which derives this complexity formally.
>
> [1] Training Data Influence Analysis and Estimation: A Survey,
>
> ---
>
> **Response to Weakness 3: The improvements are not significant**
>
> - First of all, we argue that **the results should be interpreted from a holistic perspective**. Our evaluation spans multiple tasks that require different capabilities and covers both classification and generation settings. **Our method achieves the best overall performance across all these scenarios.** This demonstrates both the applicability and the robustness of our approach.
>
> - Second, when scaling to larger models and to generation tasks (Table 3), our method outperforms baselines by a larger margin. This is particularly important given the prevalence of LLMs and generation tasks.
>
> - Finally, the goal of this work is not to propose a state-of-the-art influence estimator that dramatically outperforms existing methods. We never make such a claim. Instead, our contributions are: (1) introducing a fresh, gradient-free perspective on influence analysis based on $\Delta p$, which is conceptually different from existing approaches; and (2) empirically demonstrating that this perspective yields a strong and efficient alternative to widely-used influence estimators. We believe the experiments support this claim.
>
> We kindly ask you to re-evaluate the value of our approach in light of the above explanations and our clarification regarding its efficiency.
>
> ---
>
> **Response to Weakness 4: Analysis of the learned representations**
>
> Our analysis is intended to show that the method indeed learns meaningful influence information. In general, examples from the same class tend to exert greater mutual influence. This property is widely used to validate influence estimates, commonly referred to as class attribution [1], which checks whether highly influential examples belong to the same class.
>
> [1] Do Influence Functions Work on Large Language Models?
>
> ---
>
> **Response to Weakness 5: Why is the correlation plot necessary?**
>
> The core motivation and goal of our work is to learn influence representations that capture the influence relationships between data points in a way that the distances between representations correlate to their influence. Therefore, the correlation plot is necessary and a direct evaluation of the effectiveness of our approch, i.e., whether the these representations really capture influence.

---

> ### Author Response · Authors · 2025-11-20
> **Responses to Questions**
>
> **Why is the complexity of retraining-based approaches O(D · D · M)?**
>
> In retraining-based influence estimation, the influence of a single example is computed by retraining a model on the dataset excluding that example, which costs $O((D-1) · M)$ (with $D$ = dataset size and $M$ = number of model parameters). To compute the influence of all training examples, this procedure needs to be repeated $D$ times, leading to a total complexity of $O(D · (D-1) · M)$, which can also be written as $O(DDM)$.

---

### Official Review · Reviewer_MQvG · 2025-11-01

**Soundness:** 2
**Presentation:** 2
**Contribution:** 2
**Rating:** 2
**Confidence:** 4

**Summary:**

The paper proposes INFTRACE, a new method for estimating the influence of training data without relying on gradients or retraining. Instead, it leverages training dynamics and uses contrastive learning to distill these signals into an embedding space, where distance represents influence strength.

**Strengths:**

1. The proposed is efficient compared to gradient-based and retraining-based alternatives, providing another good zero-order data attribution method.
2. Provided comprehensive experiments across datasets and model scales.

**Weaknesses:**

1. Missing references: the beginning of the paper as well as Section 2 all mentioned most influential examples. However, the reference are incomplete given some noticeable works by a quick search, e.g., [1] and [2]. Other missing references such as various efficient alternatives (Line 44) does not include/discuss representative references such as [3, 4], and training dynamics influence such as [5, 6, 7]. Some arguments are not well-justified and existing works suggest the contrary, making poisoning this work in the literature difficult.
2. Conceptual/Factual Error: For instance Equation (1) and (2) both define $\theta' = \arg\max_{\theta} \ell$, which is confusing and should be $\arg\min$. Moreover, Equation (1) is not how the original influence function (Koh & Liang 2017) is defined.
3. Unclear motivation: The zero-order approximation of the influence function, such as TracIn [5], has already been proposed, where no training gradient or retraining is required. I see little discussion on why the contrastive learning is required while there is an easy-to-use alternative that measures what you want directly.

[1] Yuzheng Hu, Pingbang Hu, Han Zhao, and Jiaqi W. Ma. Most influential subset selection: Challenges, promises, and beyond.

[2] Huang, Jenny Y., David R. Burt, Yunyi Shen, Tin D. Nguyen, and Tamara Broderick. Approximations to worst-case data dropping: unmasking failure modes.

[3] Grosse, R., Bae, J., Anil, C., Elhage, N., Tamkin, A., Tajdini, A., Steiner, B., Li, D., Durmus, E., Perez, E. and Hubinger, E., 2023. Studying large language model generalization with influence functions.

[4] Wang, Jiachen Tianhao, Tong Wu, Dawn Song, Prateek Mittal, and Ruoxi Jia. Greats: Online selection of high-quality data for llm training in every iteration.

[5] Pruthi, Garima, Frederick Liu, Satyen Kale, and Mukund Sundararajan. Estimating training data influence by tracing gradient descent.

[6] Bae, Juhan, Wu Lin, Jonathan Lorraine, and Roger Grosse. Training data attribution via approximate unrolled differentiation.

[7] Wang, Jiachen T., Dawn Song, James Zou, Prateek Mittal, and Ruoxi Jia. Capturing the temporal dependence of training data influence.

**Questions:**

My main question is related to Weakness 1 and 3:
1. Can you provide a detailed discussion on the related literature and how your method compared to the existing works?
2. Can you elaborate why the proposed method (e.g., contrastive learning) is required, while the "ground truth" influence effect you want to measure, i.e., Equation (2), can be obtained from the training dynamics?

---

> ### Author Response · Authors · 2025-11-20
> **Responses to Weaknesses**
>
> Thanks for your feedback. Please see below our responses to the weaknesses.
>
> **Response to Weakness 1: Missing references and position of our paper in the literature**
>
> - Given the vast amount of literature in this area, it is not feasible to cite every related work.
> Therefore, we cited what we believe to be the most relevant studies, **especially the representative methods from different categories of influence analysis.**
>
> - Regarding the suggested references [1,2,3,4,6,7], we are aware of most of them.
> The reason they were not included is that they all fall under gradient-based influence estimation, for which we already cite and discuss representative work.
>
> - **As for the suggested reference [5], it is already cited in our paper and is used as a baseline in our experiments (i.e., TracIn).** We also want to clarify that although TracIn is described as a “dynamic” method, the meaning of “dynamic” in their work differs from ours: their “dynamics” refer to aggregating influence estimates across multiple training checkpoints, whereas our notion of “training dynamics” specifically refers to $\Delta p$ during training.
> **More importantly, TracIn remains gradient-based, while our approach is totally gradient-free.**
>
> - The position of our work in the literature is that, unlike most existing methods that rely on gradients to estimate influence, our approach uses only $\Delta p$. **This is a fresh perspective on approaching influence analysis. It makes our method conceptually different from prior work and, by design, more efficient than gradient-based methods and more scalable to large models.**
>
> Nevertheless, we are happy to include the suggested related studies and more in the revised version.
>
> ---
>
> **Response to Weakness 2: Explanation for EQ1**
>
> Prior work has used slightly different formulations of influence, but the core idea has always been the same: measuring how removing (or up-/down-weighting) one or more training examples affects the performance on another example (or set of examples). Our definition follows this concept. This is also the definition used in the earliest leave-one-out influence studies [1], as well as in TracIn [2]. Therefore, this is not a conceptual error. Nevertheless, we will clarify this more explicitly in the revised version. The argmax part is a typo. Thank you for pointing this out, and we will fix it.
>
> [1] Residuals and influence in regression
>
> [2] Estimating Training Data Influence by Tracing Gradient Descent
>
> ---
>
>  **Response to Weakness 3: Comparison with TracIn "for which no gradients are required"**
>
> We would like to correct one point: **TracIn is a gradient-based method**. We kindly ask you to check the TracIn paper and method description, in particular, Section 3.2 and Formula 1.
> Our method, in contrast, is entirely gradient-free and therefore fundamentally different from it.
>
> Regarding our use of contrastive learning: as explained in Lines 205–211, directly fitting a regression model to predict the exact $\Delta p$ is difficult. Instead, we use a contrastive objective to learn the relative $\Delta p$, which is more stable and tractable.

---

> ### Author Response · Authors · 2025-11-20
> **Responses to Questions**
>
> Please see below our answers to your questions.
>
> ---
>
> **Response to Q1: Relation to (and differences from) existing methods**
>
> Again, most existing methods rely on gradients to measure influence.
> Our approach, however, is gradient-free, which is a fundamental methodological difference. This is crucial because:
> - Gradient-based approaches inevitably struggle to scale to large models; although there exist efficient approximations, they still face various limitations.
>
> - Our method learns influence solely from the observed $\Delta p$ during training via representation learning. This opens a new direction for influence estimation, and more advanced methods can be developed in the future.
>
> ---
>
> **Response to Q2: Why not directly use the collected $\Delta p$**
>
> The “ground-truth” influence signal ($\Delta p$) observed during training cannot be directly used for two reasons:
>
> - It only allows us to observe the effect of each batch of training examples on the sampled validation examples in that step, whereas we aim to estimate the influence of every training point on every validation point. Therefore, we need a method to learn the patterns of observed, entangled (i.e., batch-to-single) influence, and then generalize to unobserved data pairs.
> - A secondary reason is that the $\Delta p$ observed from a single batch may be noisy due to stochasticity in minibatch updates. Therefore, we aggregate $\Delta p$ across the entire training process to cancel the noise and learn a stable influence estimate.

---

> ### Comment · Reviewer_MQvG · 2025-11-20
>
> I have one further follow-up, and I'll close the discussion since I'm now pretty confident about my overall assessment of the paper.
>
> **Weakness 1/3, and Question 1** TracIn originates from an idealized, zero-order formulation (Section 3.1, TracInIdeal), and the gradient parts are only for approximation. This is just one example of a zero-order data attribution method: for instance, there's a whole field on *simulator methods* (see [6] for a holistic survey) that does not require gradients and applies simulators (or models) to learn the ground truth influence proposed in their setup.
>
> Overall, after also looking at other rebuttals and the discussion with other reviewers, it is evident that the authors' understanding of the literature is either limited or inaccurate, validating my original judgment.
>
> [6] Deng, Junwei, et al. "A Survey of Data Attribution: Methods, Applications, and Evaluation in the Era of Generative AI." (2025).

---

> > ### Author Response · Authors · 2025-11-21
> > **Response to "TracInIdeal"**
> >
> > We respectfully ask you to revisit the full TracIn paper. Their Section 3.1 (“TracInIdeal”) simply introduces the definition of influence—the change in loss after training on a specific data point.
> >
> > However, **the key issue is not the definition itself, but how it is actually computed**. In fact, our paper adopts a similar definition (Eq. 1 and its variant Eq. 2). The essential difference lies in the estimation procedure: **TracIn estimates their defined influence using gradients, whereas our method estimates it solely from $\Delta p$, without relying on any gradient information.** This is the fundamental distinction between the two approaches.

---

> ### Comment · Reviewer_MQvG · 2025-11-21
>
> My point is that, conceptually, it doesn't really matter what TracIn (or TracInCP) does. I'm focusing on TracInIdeal, where conceptually, if you are able to trace the whole training trajectory, you are able to compute TracInIdeal exactly without any approximation (hence no gradient). If I'm not missing the point, the main reason why the TracIn's authors approximate TracInIdeal with TracIn, and then TracInCP, is that they want to avoid replaying/tracking the whole training trajectory. However, in the proposed method, I believe this (i.e., tracking the whole training trajectory) has been done exactly, and then an overhead on the simulator training on the tracked quantity ($\Delta p$) is further added. In contrast, if you can trace the whole training run, at each step, TracInIdeal only needs to track the per-sample loss (which is a scalar) within the batch, which is actually more efficient in terms of memory cost.
>
> Also, the simulator method is something the authors should recognize: there is a line of existing literature, which is all learning-based without training gradients. The authors should discuss them carefully in order to position the work in the literature.

---

> > ### Author Response · Authors · 2025-11-21
> >
> > ### **1. Comparison to TracIn**
> > No. We would like to politely point out that your understanding is incorrect.
> >
> > The “replay” process in TracIn-Ideal is **practically impossible**, because it assumes an **unrealizable training procedure**:
> > 1. batch size = 1, and
> > 2. every batch requires evaluation on the entire test (or validation) set.
> >
> > In contrast, our collection of $\Delta p$ is based on a **realistic and practical training pipeline**:
> > **we use normal batch sizes, and each batch is evaluated only on a small number of validation/test examples** (the only modification to training).
> > We also explicitly measure the additional cost of this step, which can be quantified by a factor normalized to training cost $\lambda$, and **$\lambda$ can be as small as 0.1.** (Lines 284-292, and Figure 4 middle). Therefore, collecting our “idealized” influence from training is entirely practical.
> >
> > That said, our collected $\Delta p$ is incomplete because it **only captures the influence from each batch of training data to the sampled validation data.**
> > This limitation motivates the design of InfTrace, which uses a contrastive objective to extract patterns from this limited set of observed influences, and then generalizes to predict unobserved influences.
> >
> > ---
> > ### **2. Comparison to simulator-based approaches**
> > We were aware of this line of research. However,  we consider them to be variants of retraining-based approaches: **they require retraining the model many times on different subsets and collecting the change of losses, and then fitting a simulator to map data to loss changes.**
> >
> > For example, Datamodels [1]—described as a landmark method in the survey you suggested—explicitly reports the number of retrained models in Table 1, **ranging from 600,000 to 1,500,000**, which is extremely hard to scale to large models and large datasets.
> >
> > Although we share the high-level idea of predicting unobserved influence from observed influence, **the difference is clear: InfTrace does not require any retraining.** Our contribution is that we reduce the extra training cost to as little as 0.1×, instead of the thousands or more of retrainings required by previous methods.
> >
> > Method-wise, another contribution is that we also explore how to learn a generalized pattern from such a small set of observed influence data. We found that directly fitting a regression model or surrogate model does not work (see Lines 205-211). Therefore, we propose a novel contrastive learning approach, which sidesteps the difficulty of estimating exact influence and instead learns relative influence—and still achieves strong performance. To the best of our knowledge, this has not been done in prior research.

---

> > ### Author Response · Authors · 2025-11-21
> >
> > Although we would prefer to focus on our own work rather than discussing related methods, we do feel that you have drawn an inappropriate and inaccurate comparison between our approach and TracIn—especially TracInIdeal—which has affected your understanding and assessment of our method. Therefore, we would like to clarify this point further.
> >
> > To state it more clearly: **What TracInIdeal describes is to train the model on each single data point (multiple times) and compute how that particular point changes the loss of all test examples.** This is equivalent to performing a full validation at *every* training step, with the batch size forced to be 1. Tracing the entire training run in this manner is *extremely* inefficient and in practice infeasible. It is fundamentally different from the way we trace training dynamics ($\Delta p$). Therefore, the statement that TracInIdeal is more efficient is incorrect.

---

> > > ### Comment · Reviewer_MQvG · 2025-11-21
> > >
> > > I'm fully aware of the assumption of TracInIdeal, and I consider it as a technical condition. All I'm trying to say in the above discussion is that, "conceptually", I don't think the current work recognize/discuss the existing literature enough. TracIn is just one example that "conceptually", can be gradient-free and track the training dynamics. Also, it is unfair to exclude discussion on the whole line of simulator-based methods since there exists one that happens to also be retraining-based, as there exists several non-retraining-based simulator methods.
> > >
> > > Regardless, the discussion diverges a bit from the core of the problem, i.e., why contrastive learning is required and what's the theoretical motivation behind it. I do not see a clear reason/justification in the paper.
> > >
> > > Another concern is on the experimental design, where some comparison doesn't seem fair. For instance, I imagine that when comparing with other methods, the "target function" used is not the model confidence on the validation data as the paper suggest (Eq. 2), but the validation loss instead. This is a bit concerning as this might lead to unfair comparison, as the methods you compare with are computing different target essentially.

---

> > > > ### Author Response · Authors · 2025-11-21
> > > >
> > > > Yes, we understand that you are referring to the method conceptually. However, we would like to emphasize that how a method works conceptually is less important than how it works practically. The former only provides a hypothesis or motivation, whereas the latter defines the substance of the method. Many papers in the AI literature may share similar conceptual objectives, but that does not make them redundant as long as they introduce new methodologies or improvements. Our method and TracIn are different in terms of methods—this is the point we are trying to make.
> > > >
> > > > Nevertheless, we want to stress that the above clarifications are only meant to explain that our method differs from TracIn and simulator-based approaches. We acknowledge that these distinctions were not sufficiently discussed in the paper. We appreciate you pointing out the related work, and we will incorporate your suggestions and explicitly discuss the differences between our method and other “dynamic” approaches, as well as our relation to simulator-based methods in the next revision.
> > > >
> > > > **The rationale for contrastive learning**
> > > >
> > > > Now we would like to explain why contrastive learning is needed.
> > > > Given the collected Δp, the most straightforward approach is to train a regressor to predict the exact Δp, as done in Data Model and other simulator-based methods. However, this does not work in our setting because we only collect limited Δp (since we do not frequently retrain the model), which is insufficient to fit an accurate regression model. In contrast, **contrastive learning only needs to model the relative influence relationships.** This greatly reduces the effect of noise and is a much easier task, so that it can work with limited data. To give an intuitive analogy: given two students’ previous grades, one task is to predict their exact scores in the next exam, and the other is to predict who will perform better. The latter is clearly easier, since it is entailed by the former.
> > > >
> > > > ---
> > > >
> > > > **Regarding fair comparison**
> > > >
> > > >  Δl and Δp are simply different forms of influence. Regardless of which one is used, the ultimate goal is to perform well on downstream tasks.
> > > > For example, in coreset selection, many different target signals can be used to assess data importance, but the real objective is always how much the selected data improves the model.
> > > > The other two tasks (mislabeled data identification, harmful data detection) are likewise. All three of our main evaluation tasks are Δp- or Δl-agnostic.
> > > >
> > > > In other words, **we did not conduct any circular evaluation**: the model does not directly fit the evaluation targets. Only Figure 3 evaluates directly on Δp, and that is purely for internal analysis.
> > > >
> > > > ---
> > > >
> > > > We hope the above clarifies the issue.

---

### Official Review · Reviewer_2GkH · 2025-11-03

**Soundness:** 2
**Presentation:** 2
**Contribution:** 2
**Rating:** 4
**Confidence:** 4

**Summary:**

The paper introduces INFTRACE, a data influence estimation framework that utilizes Contrastive Learning (CL) to distill influence signals from model training dynamics. INFTRACE projects influence into a low-dimensional embedding space, demonstrating strong correlation with empirical influence and showing utility in mislabeled data identification, coreset selection, and data attribution for large models.

**Strengths:**

1. The framework introduces a novel perspective by distilling influence signals directly from model training dynamics, which is both an interesting finding and a practical approach.

2. The method demonstrates strong utility across critical downstream applications, including mislabeled data detection, harmful data detection, and coreset selection, with validation across different model scales.

3. The contrastive learning formulation bypasses the computational bottlenecks of gradient-based methods, making influence estimation tractable for large-scale models.

**Weaknesses:**

1. My concern is the practical utility of coreset selection. Performance saturates at 50% of data on SNLI and SST-2, with the 10% INFTRACE coreset achieving lower accuracy than simply using 50% of data. I wonder why practitioners would prefer a 10% coreset in this case when using more data yields better performance. Also, why does saturation occur so quickly—is this a model capacity issue or data redundancy? Another concern is that evaluation is limited to in-domain test sets. I wonder if INFTRACE-selected coresets would maintain their advantages when evaluated on out-of-domain NLI tasks.

2. I wonder if the authors compared against computing influence as embedding similarity at earlier layers (e.g., word embeddings) without contrastive training. This ablation seems critical to justify whether contrastive learning offers unique value over simpler content-based similarity approaches.

3. While computational complexity is discussed, I wonder about the actual runtime. The paper would be stronger if it included wall-clock comparisons of CL training + influence calculation versus baselines across different scales to validate the efficiency claims.

4. Table 2 shows detection rates, but I'm curious whether the these amount of mislabeled data actually hurts performance. It would be helpful to report baseline accuracy on clean data and accuracy with 1%, 5%, 10% mislabeling to quantify how much degradation INFTRACE helps mitigate.

5. How sensitive is the method to the absolute |D_val| size? Also, how is the initial D_val selected from the full dataset?

6. In Figure 5, DataInf and RepSim use colors too similar to other baselines (GradDot, TracIn), making visual comparison difficult. Using more distinct colors would improve readability.

**Questions:**

Please see the Weaknesses for the details.

---

> ### Author Response · Authors · 2025-11-20
>
> Thanks for your helpful feedback. We answer your questions below.
>
> **Weakness 1: The utility of coreset selection**
>
> Coreset selection is important for low-resource settings, for reducing energy consumption, and especially for large models where training is extremely expensive. For example, a 10% subset uses five times fewer training resources compared to a 50% subset, which makes it meaningful in the abovementioned scenarios.
>
> More importantly, in our paper, coreset selection is treated as an evaluation task, so we care about the intrinsic results, i.e., whether our method outperforms the baselines under the same selection ratio. Experimental results confirm this, therefore showing the effectiveness of InfTrace.
>
> Regarding convergence behavior and cross-domain generalization, they are separate research problems and beyond the scope of this paper.
>
> ---
>
> **Weakness 2: Missing comparison with embedding-based similarity**
>
> In fact, a similar method is already included in our paper as a baseline, namely RepSim, which uses the final-layer representations to compute similarity because the final layer is expected to encode more task-relevant information. This is consistent with common practice in prior work.
>
> ---
>
> **Weakness 3: Runtime comparisons**
>
> Our method is conceptually different from prior gradient-based approaches. The additional cost in our method comes from the extra model training cost, whereas the computational cost of gradient-based methods comes from gradient computation. This means the runtime are not directly comparable because they correspond to different parts of the computation pipeline. Nevertheless, we will include a total runtime comparison for reference.
>
> ---
>
> **Weakness 4: The impact of mislabeled data on model performance**
>
> As with Question 1, mislabeled-data identification is used as an evaluation task in our paper similar to prior studies. Therefore, we care about the intrinsic results, i.e., whether our method is more accurate than baselines in detecting mislabeled examples. The effect of mislabeled data on overall model performance depends on many other factors (model, domain, task), which is a separate research problem and is outside the scope of this paper. To the best of our knowledge, prior work also does not include such discussions.
>
> ---
>
> **Weakness 5: The impact of the absolute value of $|D_{val}|$**
>
> We believe the relative ratio is a more meaningful measurement: InfTrace aims to learn the influence of all training examples; the more training examples we have, the more triplets we need, i.e., a larger $|D_{val}|$. This provides more practical guidance for applying our method, i.e., deciding the absolute $|D_{val}|$ according to their training size.
>
> ---
>
> **Weakness 6: The color choice in Figure 5**
>
> Thanks for your helpful suggestion. We will use a different color in the revised paper.

---

### Note · Authors · 2026-01-25

I have read and agree with the venue's withdrawal policy on behalf of myself and my co-authors.